# Reduction in the Sensor Effect on Acoustic Emission Data to Create a Generalizable Library by Data Merging

**DOI:** 10.3390/s24082421

**Published:** 2024-04-10

**Authors:** Xi Chen, Nathalie Godin, Aurélien Doitrand, Claudio Fusco

**Affiliations:** INSA-Lyon, Universite Claude Bernard Lyon 1, CNRS, MATEIS, UMR5510, 69621 Villeurbanne, France; nathalie.godin@insa-lyon.fr (N.G.); claudio.fusco@insa-lyon.fr (C.F.)

**Keywords:** acoustic emission, pencil-lead break test, PMMA plate, sensor effect, data merging

## Abstract

The aim of this paper is to discuss the effect of the sensor on the acoustic emission (AE) signature and to develop a methodology to reduce the sensor effect. Pencil leads are broken on PMMA plates at different source–sensor distances, and the resulting waves are detected with different sensors. Several transducers, commonly used for acoustic emission measurements, are compared with regard to their ability to reproduce the characteristic shapes of plate waves. Their consequences for AE descriptors are discussed. Their different responses show why similar test specimens and test conditions can yield disparate results. This sensor effect will furthermore make the classification of different AE sources more difficult. In this context, a specific procedure is proposed to reduce the sensor effect and to propose an efficient selection of descriptors for data merging. Principal Component Analysis has demonstrated that using the Z-score normalized descriptor data in conjunction with the Krustal–Wallis test and identifying the outliers can help reduce the sensor effect. This procedure leads to the selection of a common descriptor set with the same distribution for all sensors. These descriptors can be merged to create a library. This result opens up new outlooks for the generalization of acoustic emission signature libraries. This aspect is a key point for the development of a database for machine learning.

## 1. Introduction

Acoustic emission (AE) is widely used in the structural health monitoring field with real-time monitoring and damage identification in order to evaluate its severity. AE refers to the transient elastic waves generated by the release of energy from a material or a process within or on the surface of the material [1,2]. The use of AE is a widely applied non-destructive testing technique for monitoring and detecting various types of damage [3]. Indeed, several damage mechanisms act as acoustic emission sources, for instance, matrix cracking, fiber break, or fiber/matrix debonding in composite materials. Piezoelectric sensors are used to record acoustic waves which propagate within the material from an AE source. In AE monitoring, the sensor is the core device. It is necessary to use highly sensitive, essentially resonant elements that may modify the signal emitted by the source. These AE sensors are usually made of a piezoelectric material such as lead zirconate titanate.

A challenge in AE monitoring is to establish a clear link between the recorded AE signals and the corresponding source. The possibility of identifying the signatures of damage mechanisms is a well-established field [4,5,6,7,8,9,10,11,12,13,14,15,16]. Most of the time, the analysis of AE data is established through empirical correlations between the damage mechanism and the recorded signal. The signal is described with several descriptors calculated in the time domain, e.g., amplitude, energy, or rise time, and in the frequency domain, e.g., peak frequency or frequency centroid. Diagnosis methods consist of three main steps: (1) descriptor extraction; (2) determination of the best set of descriptors to represent the recorded data; and (3) identification of the damage using unsupervised or supervised classification approaches. The descriptor selection helps focus on the most informative data, thus reducing complexity [6]. This is crucial for improving the classification accuracy, real-time performance, and stability in AE signal detection [7]. The methodologies based on machine learning are widely used to classify different AE sources [9,10,11]. These approaches use classification algorithms to gather signals into classes depending on their descriptor values. For the unsupervised clustering [12], each class is then associated with a specific damage mechanism. The attribution of each class to a specific damage mechanism is mainly based on empirical approaches, and the validation of this labeling remains difficult and is still a challenge. For the supervised approach [13,14], the quality of the library is crucial. A robust dataset from the AE experiments is necessary to improve the accuracy of machine learning [15,16].

AE signals and descriptors depend on several factors: (1) the damage source; (2) the specimen geometry; (3) the material properties, and (4) the type of sensor and acquisition chain. Several studies have investigated the influence of these factors on the AE signal characteristics [17,18,19,20]. The calibration curve, illustrating the sensor frequency response, strongly depends on the sensor type [21,22,23]. Godin et al. [21] compared the descriptor patterns obtained using different sensors, derived from the tensile tests of fiber-reinforced composites. A relative difference of the descriptor value larger than 20% was obtained between the two different types of sensors. Different sensors with different sensitivity may be better at identifying various AE sources [22,23]. Some authors [24,25,26,27,28] used numerical simulation to quantify the effect of the sensor on the acoustic signature. The finite element modeling of the two AE sources located at different depths from the sample surface showed that the different variations of the descriptor values were deduced using two kinds of sensors [24]. Larger sensors cannot detect signals evenly across their entire surface so that there is a variation in the waveform peak and shift in the dominant frequency [25,26,27,28]. Sause et al. [25] and Mu et al. [26] focused on the effect of the radius of the sensor tip. The difference in the response sensitivity to the frequency of the AE sensor appeared to be the main reason for the observed difference in the waveform peak and shift of the dominant frequency. The coupling medium also represents another factor in the uncertainty of recorded AE results. According to Zhang et al. [27], the coupling thickness can alter peak frequency. According to previous studies, the influence of different factors on AE characterization has been extensively studied on a laboratory scale. These works summarize some results from the literature concerning the identification of the acoustic signatures for several damage mechanisms in composite materials (matrix cracking, fiber/matrix debonding, delamination, fiber breakage, fiber pullout) from a single parameter such as average frequency [29] or amplitude and peak frequency [30]. The aforementioned works underscore a tendency concerning organic matrix composites: matrix cracking is linked to low-frequency signals, while fiber ruptures are connected with higher-frequency signals. However, notable differences are noted for the same composite in the reported results, which can be attributed to several factors including loading conditions, sensor types, and so on. Indeed, the type of sensor plays an important role in the characteristics of the recorded signals, and the results reported in [21,31] show the data recorded by both sensors (Micro80 and PicoHF) on the same specimen during the tensile test on composite materials. For example, for the glass fibers/PA6.6 polyamide composites, the mean value of the peak frequency is equal to 244 kHz for the Micro80 sensor and 578 kHz for the PicoHF sensor. The sensor noticeably distorts the AE signals. To obtain a generalizable acoustic signature for a given material, it is important to overcome the sensor effect. Because of the high sensitivity required of these sensors, it is impossible to change the sensor type. Wide-band sensor adoption will only spread when high-energy mechanisms are detected, but they are less sensitive.

However, little work has been performed in order to reduce the sensor effect. For the case where multiple sensors are used to record AE signals during mechanic tests, Guel et al. [22] proposed an approach to merge the data from two different AE sensors in the classification process. Combining features extracted from different sensors and integrating them into single features enhances the diagnostic phase and reduces the uncertainty. However, the developed approach requires the same test to be instrumented with both types of sensors. Additionally, to ensure the quality of the dataset in the machine learning of the clustering algorithm, Hamam et al. [32] simulated the performance of several sensors in recording AE signals due to fiber break. The numerical method for simulating AE signals enables the extension of training datasets for the machine learning algorithm as a step towards quantitative analysis.

There is still a challenge in suggesting the most appropriate features to describe AE signals while getting rid of the sensor effect. The main purpose of this study is to determine a procedure for combining multi-sensor data by merging data from different sensors. In this case, sensors measure the same AE sources. The sensor configuration is complementary, and the sensors do not depend on each other but they can be combined in order to give a more complete image of the AE source in the library. The main objective is to propose a method to reduce the sensor effect. The study is conducted on plates since plate-like structures are omnipresent in different fields with AE applications. The acoustic emission sources are generated by the pencil-lead break (PLB) procedure. The PLB is a common approach for generating a reproducible artificial acoustic emission (AE) source, referred to as the Hsu Nielsen source. The basic idea behind PLB tests is to break a pencil lead on a specimen surface under controlled conditions to emit a wave that sensors can then record. The emitted frequency content of this source can be controlled by the length and orientation angle of the pencil lead break [33,34]. First, this paper studies the influence of the sensor effect with different source–sensor distances on the characteristics of the AE signals obtained by PLB on a 3 mm thick PMMA plate. The experimental setup is presented in Section 2. Then, by comparing the AE results obtained by five different sensors, we illustrate the influence of the sensor effect. A procedure is then proposed to reduce the sensor effect based on the normalized data, the Kruskal–Wallis test, outliers computation, and principal component analysis. Feature selection aims to choose a subset of features that can be used for all sensors. Concluding remarks are provided in Section 4.

## 2. Experimental Setup and AE Data Post-Processing

### 2.1. Experimental Setup

We conducted a set of PLB tests on a 3 mm thick square plate (1 m × 1 m) of polymethyl methacrylate (PMMA). Figure 1 shows the experimental setup and the position of the sensor at the plate surface center (red point in Figure 1).

The artificial source is imposed by a pencil-lead breakage on the plate surface using a pencil lead with 2H hardness, 0.5 mm diameter, and 4 mm length, the contact angle between the lead and the specimen being $45^{\circ }$. The PLB tests are performed on the specimen at 12 positions along the center line (the blue points in Figure 1) with increasing source–sensor distances, from 20 mm (close to the sensor) to 450 mm (far away from the sensor). The PLB tests performed at different locations from the sensors allow the characterization of the AE signature of the same source including the influence of the wave propagation effect. In one test batch, we set the same experimental setting, and we repeat PLB six times at every position. This procedure allows us to assess repeatability. Between two different test batches, the sensor is removed and re-placed in the same position. We carry out six batches of PLB tests with one sensor on the same specimen so that 36 signals are recorded at every position. Vacuum grease is used to couple the sensor bottom surface to the PMMA plate.

AE monitoring was conducted using five different types of sensors (PKBBI sensor, WD sensor, Micro200HF sensor, Nano30 sensor, and Micro80 sensor (Mistras Group, Princeton, NJ, USA)). Except for the PKBBI sensor, other sensors are connected to a preamplifier (40 dB gain, type 20H). The PKBBI sensor is linked to a preamplifier that has a 26 dB preamplification and a Zener barrier of 14 dB that lowers the output voltage to 5 V, which is equivalent to other sensors’ preamplification (40 dB). The pre-filter is set from 20 kHz to 2 MHz, and all signals are recorded at a 5 MHz sampling rate by a MISTRAS system. The acquisition threshold is set at 40 dB. The acquisition parameters, the peak definition time (PDT), the hit definition time (HDT), and the hit lockout time (HLT) are, respectively, 50 μs, 100 μs, and 600 μs.

Table 1 summarizes the main characteristics of the sensors. Figure 2 shows the sensitivity curves for two sensors (Nano30 and PKBBI) provided by the manufacturer. The data demonstrate that the sensitivity of each sensor varies across different frequencies. These different sensors have been selected for their different and complementary frequency responses (Table 1). In the sequel, we investigated the influence of these different responses in the frequency domain on recorded AE signals.

A method based on the waveform’s correlation coefficient (CC) in the time domain or the frequency domain is used to evaluate the repeatability of AE results. By evaluating the CC between two randomly selected signals, on the one hand, the mean CC in each batch has to be larger than 0.8; on the other hand, the mean CC between batches has to be larger than 0.5. In case a recorded signal does not fulfill these two requirements, it is removed from the dataset as the obtained difference may originate from an experimental bias.

### 2.2. Excited Waves

The thickness of the specimen is much smaller than the other two dimensions. In this case, AE sources generate dispersive Lamb waves. The extensional mode has its largest displacement components in the plane of the plate, while the flexural mode has its largest displacement component perpendicular to the plate. The sensors, attached to the face of the plate, are mainly sensitive to the out-of-plane component of any propagating waves. This artificial source on the top surface is shown to be equal to an out-of-plane monopole [37] that induces a high amplitude $A_{0}$ component and a low amplitude $S_{0}$ component.

The group velocity curves representing the variation in wave propagation velocities with frequency are shown in Figure 3. This is a theory solution obtained by the GUIGUW (Graphical User Interface for Guided Ultrasonic Waves) assuming an isotropic plate. A plate of 3 mm allows the excitation of $A_{1}$, $A_{2}$, and $S_{1}$ modes in addition to the fundamental modes within the frequency range of up to 1 MHz. Nevertheless, for this plate thickness, only the zero-order symmetric mode ($S_{0}$) and the anti-symmetric mode ($A_{0}$) exist at a frequency under 180 kHz. At low frequencies, the group velocities of the two modes are significantly different, and after a suitable propagation distance, the arrivals of these modes should appear well separated in the time domain.

### 2.3. Dataset Description and Data Post-Processing

The signals were post-processed with MATLAB following the approach in [38] to calculate the descriptors derived from AE signals. Firstly, a pre-trigger removing was used to delete the part before the beginning of signals, and an energy criterion was applied to determine the end of signals. For each point in the waveform, the cumulative energy (Equation (Equation 1)) computed from the beginning is compared to the energy contained in a 10 μs length window following that point. If this energy is less than a threshold equal to 0.01 % of the cumulative energy, then the corresponding point represents the end of the signal.
(1)$$E\left(t\right)=\int_{0}^{t}V{\left(t\right)}^{2}dt$$

The different descriptors of AE signals are extracted from the signal. The fast Fourier transform (FFT) of each signal is also calculated to describe its frequency content. Finally, up to 23 descriptors in the time and frequency domains are calculated. Table 2 summarizes the main descriptors used in this study. Here, the peak frequency is the frequency value where the magnitude of the FFT is the largest over the frequency range of 0–1000 kHz. To quantify the sensor effect, two other frequency peaks are introduced, noted $PF_{1}$ and $PF_{2}$. $PF_{1}$ indicates the peak frequency over the frequency range below 100 kHz, and $PF_{2}$ indicates that over the frequency range above 100 kHz, as shown in Figure 4.

The signals are analyzed in the time domain and the frequency domain with the short-time Fourier transform (STFT) as well as in the time–frequency domain.

### 2.4. Procedure to Reduce Sensor Effect towards Common Set of Descriptors

This section presents the procedure developed to reduce the sensor effect and to select descriptors to be merged to create a library. The main steps for describing the feature selection methods are as follows:(1)Raw descriptor sets are normalized;(2)The Kruskal–Wallis (KW) test is applied to normalized descriptors to evaluate the normalization method and whether descriptors come from the same distribution;(3)Descriptors with too many outliers are removed and, for the other descriptors, the signal outliers are removed from the dataset;(4)Principal component analysis (PCA) is applied to the remaining normalized descriptors and datasets.

The different steps are presented in detail hereafter.

#### 2.4.1. Normalization

Data normalization is a common preprocessing approach to ensure that each feature contributes equally to the outcome of the classification algorithms, which leads to the better generalization in predictive models [39]. Several normalization techniques are compared in the paper to overcome the sensor effect so that responses measured by different sensors can be quantitatively analyzed and compared. About ten different normalization methods, mentioned in [39], are used to normalize the AE results. Among these methods, the Z-score normalization (ZSN) method considers the mean value and the standard deviation of the dataset to obtain standardized data, having a mean of zero and a standard deviation of one:(2)$$x_{i,n}^{\prime }=\frac{x_{i,n}-\mu_{i}}{\sigma_{i}},$$
where $x_{i,n}$ is the *n*-th value of the *i*-th descriptor; $x_{i,n}^{\prime }$ is the new value after normalization; $\mu_{i}$ and $\sigma_{i}$ are the mean and the standard deviation of the *i*-th descriptor. The empirical cumulative distribution function (ECDF) is a statistical tool used to estimate the probability of a variable falling within particular ranges by describing the cumulative probability distribution of a group of data points. The ECDF of each descriptor recorded by five sensors, derived from the raw data and the normalized data, is analyzed. After normalization, it is possible to compare the responses obtained from AE sensors for each descriptor.

#### 2.4.2. Kruskal–Wallis Test

The KW test is performed to (1) determine the best way to normalize the data, and (2) select the descriptors coming from the same distribution for different sensors. This test is non-parametric. Each feature or descriptor for each normalization method is evaluated individually. This allows one to test the null hypothesis $H_{0}$ that five independent datasets originate from the same distribution. The KW test is a robust and straightforward statistical test comparing the average ranks of all data points for each descriptor to determine whether there are statistically significant differences [40]. We set the probability α to 0.05. All datasets come from the same distribution when the KW test’s *p*-value is greater than α. As the calculated probability value *p* is smaller than 0.05, we reject the null hypothesis at the significance level α=0.05, i.e., the difference between the five datasets is statistically significant.

#### 2.4.3. Outliers

To avoid the influence of the outliers on the classification, the third step is to exclude the selected descriptors with more than 20 outliers from the subsequent analysis. Meanwhile, for the other selected descriptors with fewer than 20 anomalies, the signal outliers are removed. The outlier is a value larger than 1.5 times the interquartile range above or below the interquartile values in the boxplot.

#### 2.4.4. Principal Component Analysis

The PCA is a multivariate data reduction technique that can reduce the multi- dimensionality without losing much information in the data [41]. It is commonly employed to reduce an enormous amount of variables by transforming it into a new set of variables that summarize the original essential patterns [42]. It is also used in multivariate data analysis to explore the first several components that contain the majority of the variation, making it easier to classify different types of damage [8,43,44]. The PCA finds patterns in data $X=[X_{1},X_{2},...]$ and expresses them as a set of new, uncorrelated variables called principal components $A_{k}$. It starts by identifying the direction in which the data varies the most, making this the first principal component. Then, it finds the next direction of maximum variance that is orthogonal to the first, which becomes the second principal component. By calculating the eigenvalue and eigenvectors from the covariance matrix $XX^{T}$, this process continues until the number of principal components is equal to the number of original variables. By focusing on the main feature matrix $A_{k}$ with high cumulative variance, the PCA helps highlight the patterns and similarities in the data. By applying the formula $Y=X\times A_{k}$ to the original data vector *X*, a new data vector is defined by the eigenvector matrix $A_{k}$[43]. To evaluate the sensor effect in the application of the PCA, one eigenvector matrix $A_{k}$ of one sensor is chosen, and the dataset recorded by other sensors is transformed by this eigenvector matrix $A_{k}$. If the new datasets of all sensors have a similarity in the PCA plan, it is proven that different AE results can be used for the library.

## 3. Results and Discussion

### 3.1. Sensor Effect on the Waveform

In this section, the influence of the sensor type on the AE response is described in terms of the waveform in the time domain and the frequency domain.

**Time domain**: Figure 5 shows the normalized signals recorded by five sensors on a 3 mm thick plate at distances of 20 mm (closest to the sensor), 100 mm, and 450 mm (far away from the sensor), respectively. Two types of wave modes moving at different velocities can be identified: $S_{0}$ and $A_{0}$ modes. These two modes are labeled in Figure 5b,c. As expected from the group velocities curves, the $S_{0}$ mode is observed at the beginning of all the signals. When the source–sensor distance is small (20 mm), it is hard to identify two different modes from waveforms in the time domain. Only the waveforms recorded by the PKBBI sensor and the WD sensor are similar.

**Frequency domain**: Figure 6 shows the corresponding normalized FFTs. The pencil lead breakage on the PMMA plate can be attributed to a low-frequency source. The primary characteristic of the FFT obtained from the PKBBI, the WD, and the Micro200HF sensors is a low-frequency characteristic, particularly below 200 kHz, as shown in Figure 6. As the propagation distance increases, the magnitude of the high-frequency range surpassing 100 kHz diminishes.

**Time–frequency analysis**: As seen from the time–frequency maps in Figure 7, the peak amplitude of $S_{0}$ mode has a frequency component of 100–500 kHz, at which it is non-dispersive. The peak amplitude of $A_{0}$ mode has a frequency mode below 100 kHz at which it is dispersive. $A_{0}$ mode dominates for the PKBBI, the WD, and the Micro200HF sensors, as shown in Figure 7a–c. For the PKKBI sensor, the $S_{0}$ mode arrives first. Two similar arrival modes, $S_{0}$ and $A_{0}$, are recorded by the WD sensor. For the Micro200HF sensor, there is a less pronounced mode $S_{0}$. Then, the $S_{0}$ mode arrives also first but it is dominant in the signals from the Micro80 sensor and the Nano30 sensor. For these two sensors, the ratio of the magnitudes $S_{0}/A_{0}$ is higher compared to the other sensors. Additionally, as observed from the mode $A_{0}$, there is a small attenuation of the mode recorded by the PKBBI and the WD sensors, where the low-frequency content continues after 200 μs. For the Micro200HF, the Micro80, and the Nano30 sensors, the low-frequency content disappears until the duration is larger than 200 μs.

As observed from these waveforms in the time domain and the frequency domain, the results demonstrate a significant effect of the sensor on the AE signature.

### 3.2. Sensor Effect on Descriptors

**Amplitude and frequency centroid**: Figure 8 shows the whole data of the amplitude and the frequency centroid recorded by five different sensors. Among the signals captured by these five different sensors, the Micro200HF sensor has the shortest amplitude variation along the 450 mm source–sensor distance and the lowest frequency centroid. The Micro80 sensor recorded the largest amplitude variation and the highest frequency centroid since it is highly sensitive above 200 kHz.

**Peak frequency**:
 **(a)** **Strong dependence on the sensor–source distance**: As demonstrated in Figure 9a, the peak frequency recorded by the Micro80 and Nano30 sensors jumps from above 100 kHz to below 100 kHz due to the different amplification of the $S_{0}$ mode. These two sensors record peak frequencies that are higher than 200 kHz when the sensor is close to the source. With increasing source–sensor distances, the peak frequency lies below 100 kHz. Figure 10 shows the peak frequency 1 and the peak frequency 2 as a function of the source–sensor distance for the Micro80 sensor. The peak frequency 1 is constant over all source–sensor distances. However, the peak frequency 2 is about 300 kHz for the source–sensor distance smaller than 100 mm. Due to an attenuation of the $S_{0}$ mode with increasing distances, especially the high frequency content above 100 kHz, the peak frequency 2 decreases to about 100 kHz when the sensor is far away from the source. The peak frequency data points recorded by the Micro80 sensor as a function of the source–sensor distance are also in Figure 10. It can be seen that, for small distances, the peak frequency is equal to the peak frequency 2, which is higher than 100 kHz. It is seen from the FFT recorded by the Micro80 sensor at a 20 mm distance (Figure 6a). Due to a decreasing normalized magnitude of the peak frequency 2 with increasing distances, the magnitudes of these two peak frequencies are close, as seen in Figure 6b. Then, the peak frequency can be one of the two frequency values at distances of about 100–150 mm. For a large source–sensor distance, as seen from the FFT recorded by the Micro80 sensor at 450 mm distance in Figure 6c, the peak–frequency is equal to the peak frequency of 1. **(b)** **Low dependence on the source–sensor distance**: For the PKBBI sensor and the Micro200HF sensor, the peak frequency is constant over wave propagation. The WD sensor recorded two different peak frequency values, but they are both lower than 100 kHz, as seen in Figure 9b. Thus, the peak frequency recorded by the PKBBI sensor and the Micro200HF sensor is independent of the source–sensor distance, and that recorded by the WD sensor is less dependent on propagation.

**Rise time**:
 **(a)** **Dependence on the arrival of the $S_{0}$ mode**: A jump in the rise time that only occurs for the Micro80 sensor is the following significant difference. The whole data of the rise time versus the amplitude recorded by different sensors is shown in Figure 11. The more transparent the marker, the farther away from the source. We can see that the closer to the source, the smaller the rise time on most data points. However, for the Micro80 sensor, some rise time values are constant, about 20 μs when the source–sensor distance is smaller than 300 mm. As it can be seen in the time–frequency map (Figure 7d) at a 100 mm distance, the $A_{0}$ mode arrives at 20 μs, but the magnitude of the $S_{0}$ mode is larger than that of the $A_{0}$ mode. Thus, the rise time of the Micro80 sensor depends on the arrival of the $S_{0}$ mode for the source–sensor distance smaller than 300 mm. Due to the same attenuation of modes on the same medium but different for different modes [45], the magnitudes of modes decrease for all sensors, and the magnitude of the $S_{0}$ mode will be smaller than the magnitude of the $A_{0}$ mode for the large source–sensor distance. Figure 12 shows two waveforms in the time domain recorded by the Micro80 sensor at 300 mm and 350 mm distances, where RT1 and RT2 denote two rise times of these two signals. We can see that the rise time RT1 at a 300 mm distance is lower than 50 μs, which contains a 25 μs pre-trigger. Whereas the rise time at a 350 mm distance is about 200 μs, thus yielding a rise time jump, although the source–sensor distance only increases by 50 mm. **(b)** **Dependence on the arrival of the $A_{0}$ mode**: The Nano30 sensor exhibits a high magnitude ratio $S_{0}/A_{0}$, similar to the Micro80 sensor. The waveform recorded by the Nano30 sensor from Figure 5b demonstrates that, despite an amplification of the $S_{0}$ mode, the magnitude of the $S_{0}$ mode is still lower than the magnitude of the $A_{0}$ mode. Due to the attenuation of modes, the amplitude of the $S_{0}$ mode is relatively small, so the rise time recorded by the Nano30 sensor is not dependent on the $S_{0}$ mode but on the $A_{0}$ mode. For other sensors, as shown in Figure 7, the magnitude of the $S_{0}$ mode is smaller than that of the $A_{0}$ mode when the distance is larger than 100 mm. Thus, the rise time recorded by these sensors depends on the arrival of the $A_{0}$ mode and increases with increasing source–sensor distances.

The analysis of the sensor effect in Section 3.1 and Section 3.2 shows that there is a non-negligible influence of the sensor types. The difference in the waveforms recorded by the different sensors results in a shift in the descriptors’ values in the time domain and the frequency domain.

### 3.3. Feature Selection to Obtain a Common Set

The ECDF of the amplitude recorded by five sensors derived from (a) the raw data and (b) the normalized data using the ZSN approach is shown in Figure 13. The curves corresponding to the raw data recorded by five sensors have a huge disparity, as seen in Figure 13a; nevertheless, these distribution functions are considerably comparable after Z-score normalization, as seen in Figure 13b. Figure 14 shows the distribution obtained for the peak frequency, and in this case, the distributions obtained appear quite different even after normalization. For the different methods and all descriptors, a visual analysis is not sufficient to determine the best normalization. In the second step, the KW test is applied to define the best way to normalize the data and to make a first selection of the pertinent descriptors that can be kept to describe the normalized data recorded by any of the sensors.

The *p*-value results of the KW test for 10 different normalization methods are shown in Table 3. The acronym signification of each method is also given in Table 3. It is shown that, if α value is 0.05, almost all the results of the MMN and MMADN are equal to zero, and the results of the ZSN, VTN, LS, and HT are the same, only three *p*-values of the descriptors smaller than α. This is because the VTN, LS, and HT all are a transformation of the ZSN. They use the ZSN results as a basis. Thus, the ZSN is a simple method with a good performance. It works better for the whole set of features because it keeps the original distribution of features and does not assume a limited range. It is also easier to use and better at dealing with outliers and making sure that each feature has an equal contribution [16]. Among the 21 descriptors calculated after normalization, only 4 are rejected as not belonging to the same distribution: Energy, Temporal Decrease, Partial Power 4, and Peak Frequency.

To avoid the bias of the outliers on the classification, the third step is to remove the outliers. Figure 15 shows the outliers, noted by the red symbol, of the normalized values of each descriptor recorded by the PKBBI sensor. The numbers in the x-axis corresponds to the order of the descriptor in Table 2. Crossed descriptors containing more than 20 anomalies will be excluded from the subsequent analysis, for instance, the roll-on frequency. The blue box indicates that all signal outliers of partial power 2, spectral slope, and spectral skurtosis to peak for the PKBBI sensor will be removed.

To date, a decision has been made to exclude certain descriptors from the subsequent analysis. These descriptors are those that are from different distributions due to the KW test and those that have more than 20 outliers. Thus, it is possible to define a common set of descriptors for all sensors: amplitude, zero-crossing rate, rise time, temporal centroid, partial power 1, partial power 2, partial power 3, frequency centroid, spectral skewness, spectral kurtosis, spectral skewness to peak and spectral skurtosis to peak—thus 12 descriptors.

### 3.4. Principal Component Analysis

We input the new dataset and the common set of descriptors into the PCA to illustrate the ability of this method to reduce the sensor effect. The principal components of each sensor are calculated, and the dataset is calculated using the eigenvector matrix, which consists of the first few primary components with the highest cumulative variance. Once one eigenvector matrix has been chosen as the reference basis, all the datasets of different sensors are projected to this reference basis. For different sensors, the percentage variability of the first principal component for five sensors is about 70%–80%, which contains much more information than one of the 12 selected components. In this article, the total of features with over 95% cumulative variance is selected. Thus, the first four principal components are selected for each sensor.

Figure 16 shows five 2D representations of the new data sets from the first two principal components (PC1, PC2) of five sensors. These two features contain about 90% information of the 12 selected components. The reference sensor is chosen as the (a) PKBBI sensor; (b) WD sensor; (c) Micro200HF sensor; (d) Micro80 sensor; and (e) Nano30 sensor, respectively. It is found that whatever the choice of the reference sensor, the data distributions of the five sensors are similar, which could be assigned to the same group.

## 4. Conclusions

Acoustic emission analysis of PLB tests highlights that the type of sensor used has a significant influence on the recorded AE signal characteristics. We can draw the following conclusions:1.While selecting the sensor does not have a significant impact on the signal detection, it does play a crucial role in determining the nature of the source. Because of the different amplification of wave modes, when using different sensors, the waveforms in the time and frequency domains differ.2.The Micro80 sensor and the Nano30 sensor can add biased information, such as the low rise time recorded by the Micro80 sensor when it is close to the source and the peak frequency jump from above 100 kHz that these two sensors record. This may result in the creation of an additional class during the process of classifying the data. The peak frequency, which is highly dependent on both the distance and sensor type, is not a reliable descriptor for classification.3.The flow chart of the detailed steps is shown in Figure 17. This procedure allows one to obtain a compact representation of the dataset with good generalization ability. The Z-score normalized method has been proven to be a simple and effective way to obtain more descriptor sets from the same distribution of sensors. After the Kruskal–Wallis test and the outlier computation, it is possible to choose a sufficiently large subset of features in order to describe the same source recorded by different sensors. This subset of features includes descriptors in both the time and frequency domains. The principal component analysis reduces the feature set’s large dimension, and the same source is clustered by the dataset recorded by multiple sensors, regardless of the reference eigenvector matrix.4.In the future, the influence of the plate thickness should be considered in the process of reducing the sensor effect. This post-processing will be applied to the PLB test with thicker plates.

## Figures and Tables

**Figure 1 sensors-24-02421-f001:**
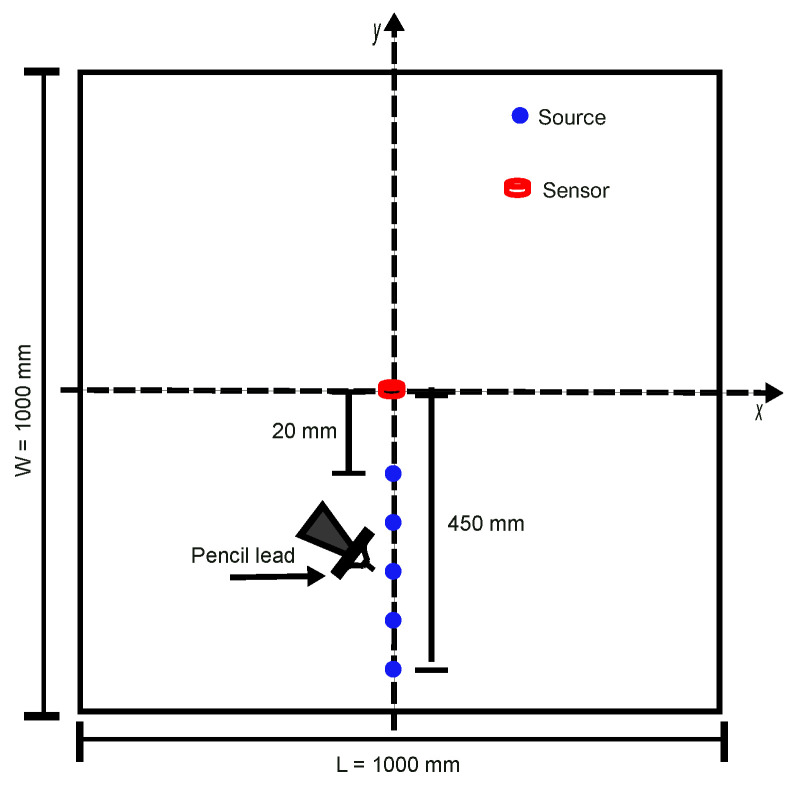
Experimental setup of PLB test performed on a 3 mm thick PMMA plate with a sensor located at the plate surface center.

**Figure 2 sensors-24-02421-f002:**
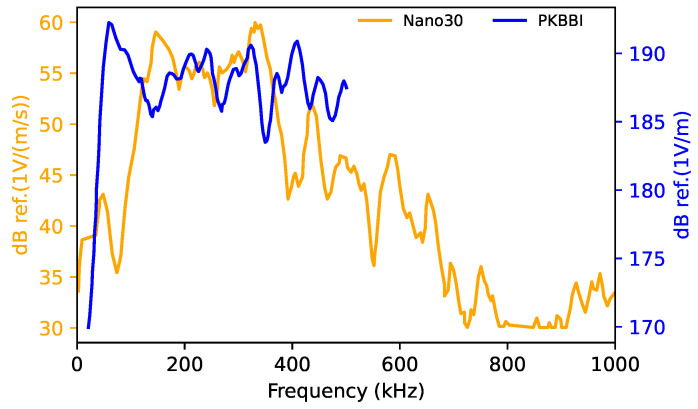
Sensitivity in the reception for the Nano30 sensor and the PKBBI sensor produced by MISTRAS Group.

**Figure 3 sensors-24-02421-f003:**
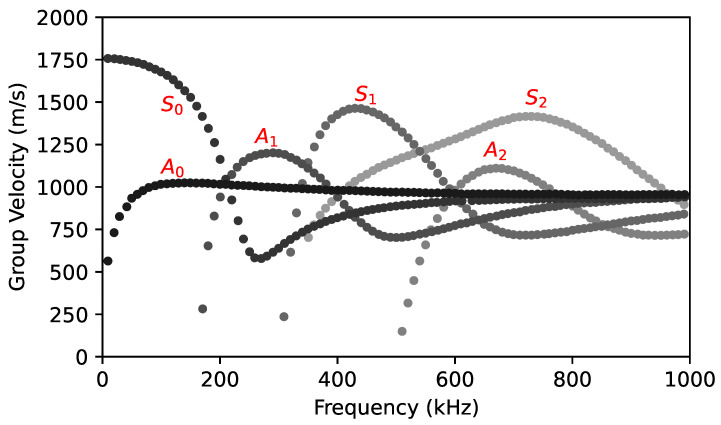
The group velocity as a function of frequency in a 3 mm thick PMMA plate.

**Figure 4 sensors-24-02421-f004:**
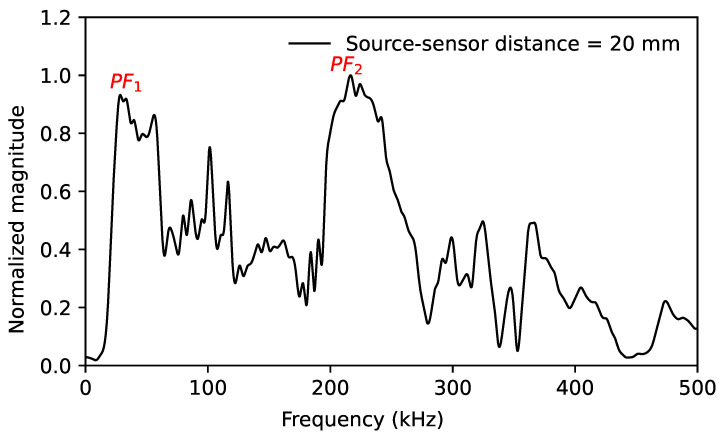
The signal from the Micro80 sensor at a distance of 20 mm illustrates two peak frequencies, $PF_{1}$ and $PF_{2}$.

**Figure 5 sensors-24-02421-f005:**
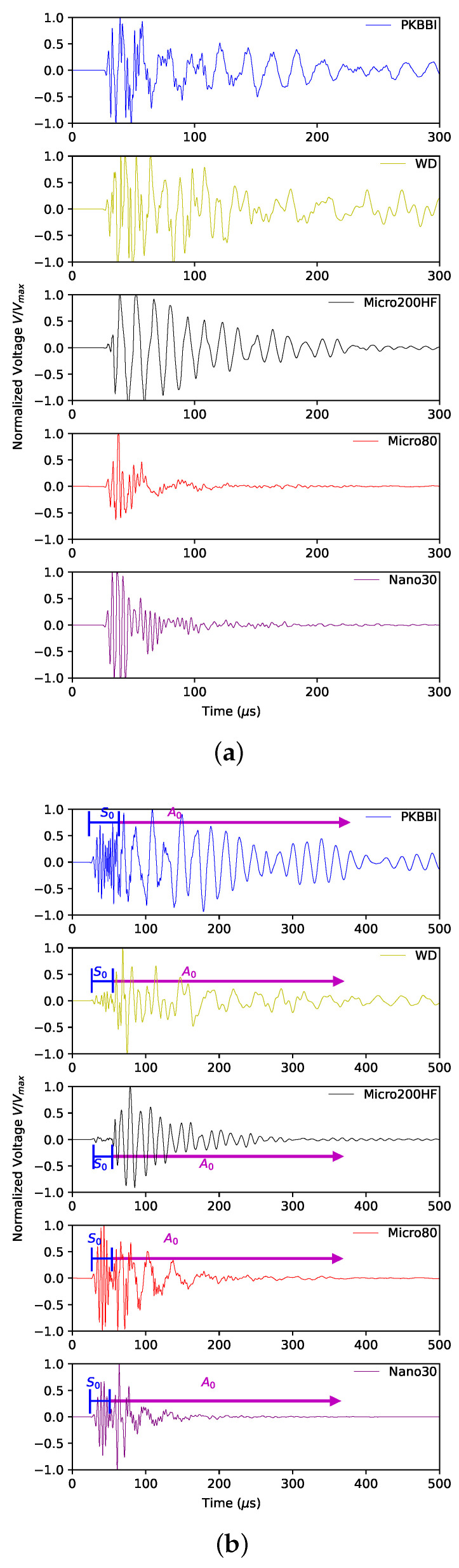
Waveforms of signals recorded by PKBBI sensor, WD sensor, Micro200HF sensor, Micro80 sensor, and Nano30 sensor on a plate at a distance of (**a**) 20 mm; (**b**) 100 mm; and (**c**) 450 mm.

**Figure 6 sensors-24-02421-f006:**
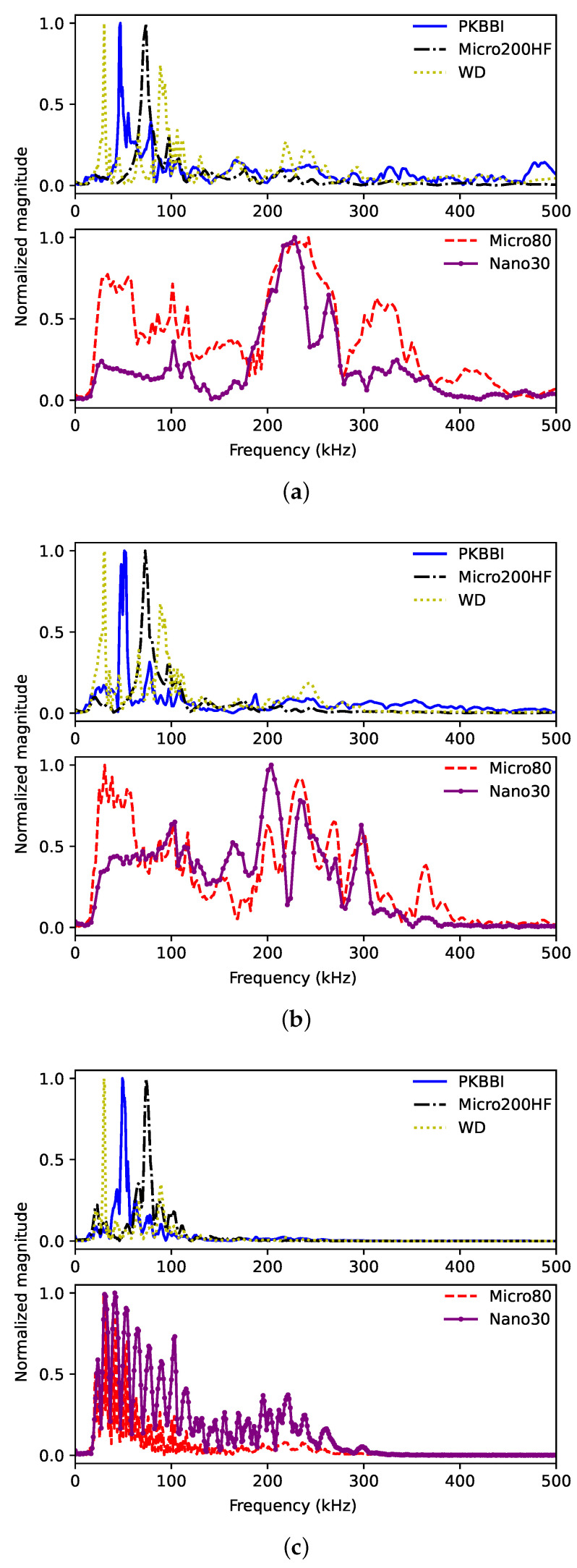
FFT of signals recorded by PKBBI sensor, WD sensor, Micro200HF sensor, Micro80 sensor, and Nano30 sensor on a plate at a distance of (**a**) 20 mm; (**b**) 100 mm; and (**c**) 450 mm.

**Figure 7 sensors-24-02421-f007:**
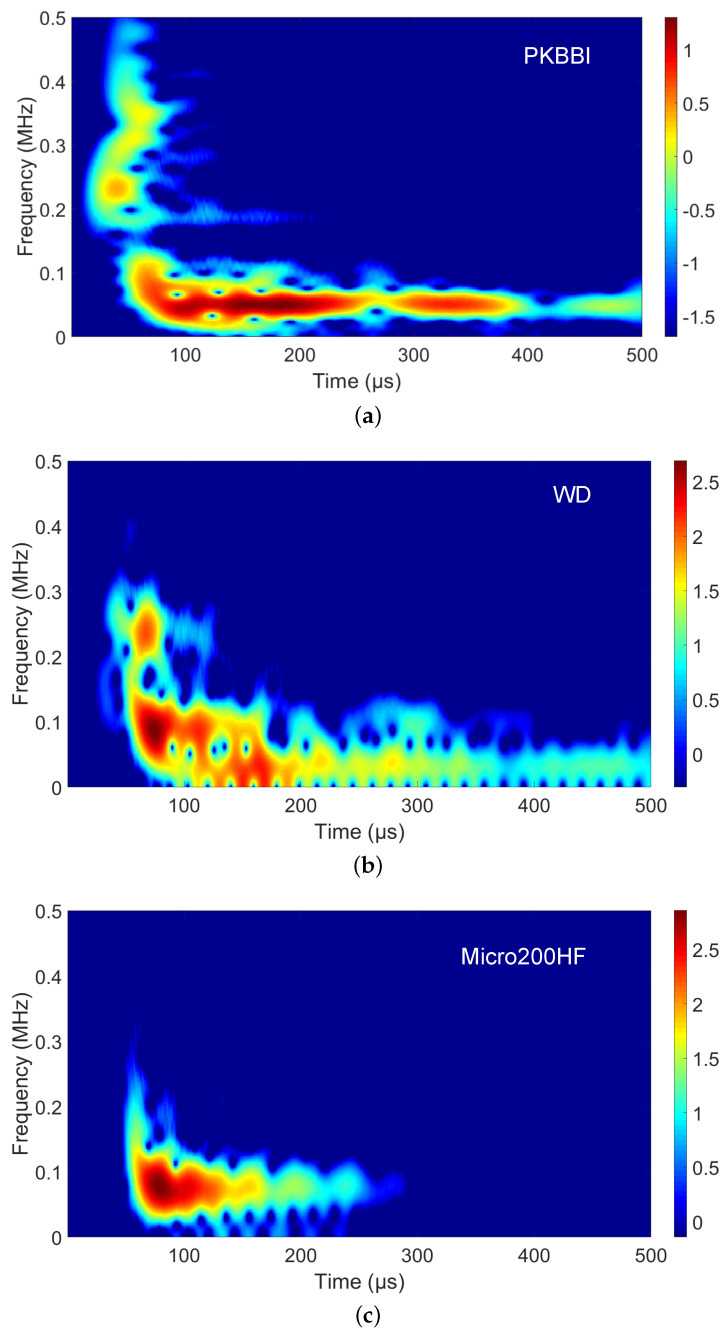
Time–frequency maps recorded by (**a**) the PKBBI sensor, (**b**) the WD sensor, (**c**) the Micro200HF sensor, (**d**) the Micro80 sensor, and (**e**) the Nano30 sensor at a 100 mm distance from the source. They correspond to the signal as shown in Figure 5.

**Figure 8 sensors-24-02421-f008:**
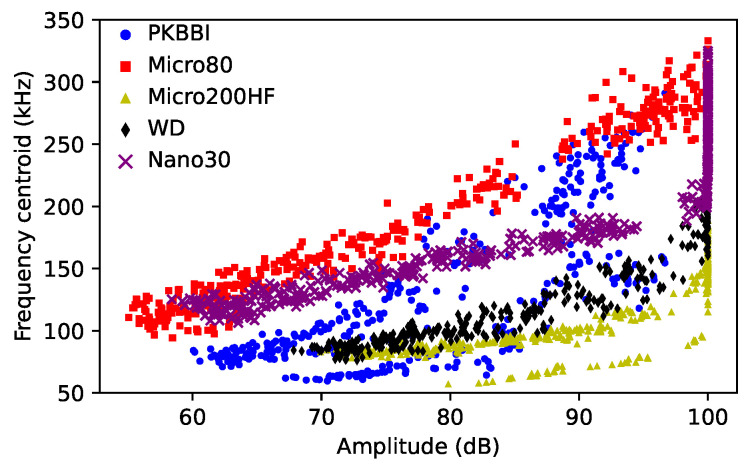
The frequency centroid versus the amplitude recorded by different sensors.

**Figure 9 sensors-24-02421-f009:**
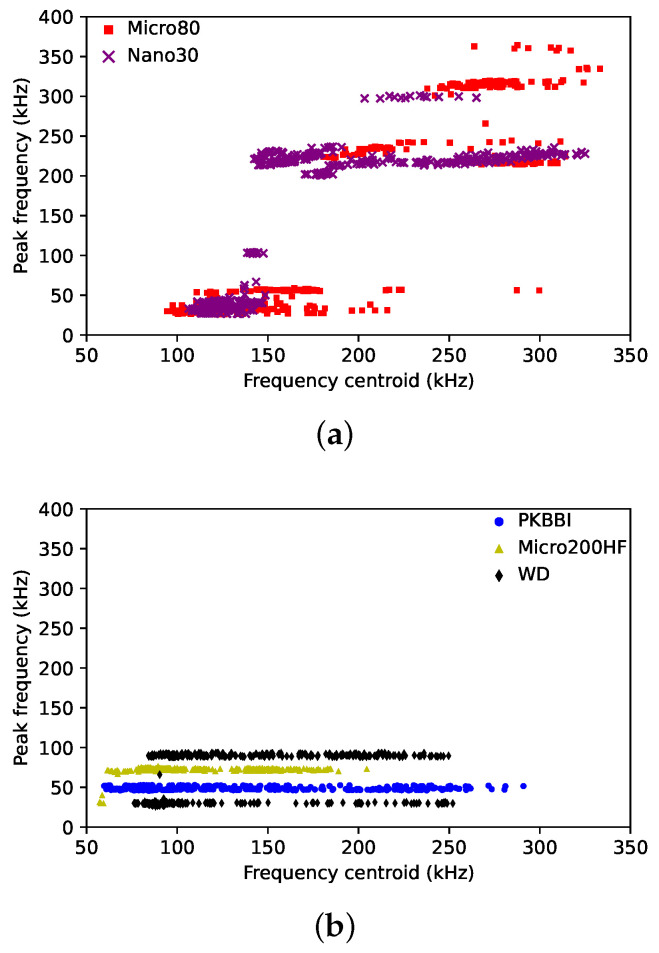
Peak frequency versus the frequency centroid recorded by (**a**) the Nano30 sensor, the Micro80 sensor, and (**b**) the PKBBI sensor, the MCIRO200HF sensor, and the WD sensor.

**Figure 10 sensors-24-02421-f010:**
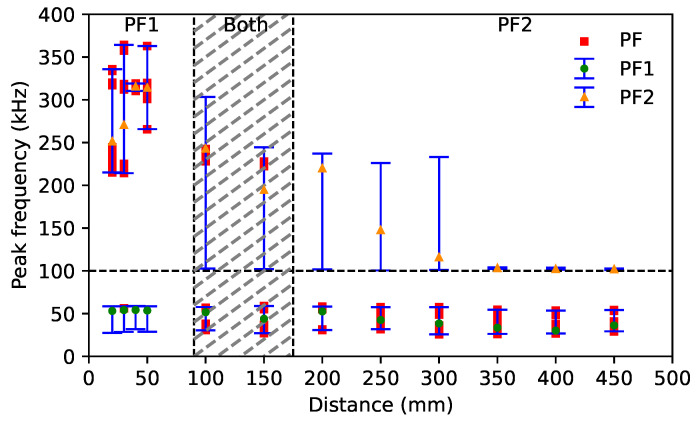
PF1 and PF2 recorded by the Micro80 sensor as a function of the source–sensor distance. PF indicates the data point of all signals. PF1 and PF2 indicate the variation range with the average value at each source–sensor distance.

**Figure 11 sensors-24-02421-f011:**
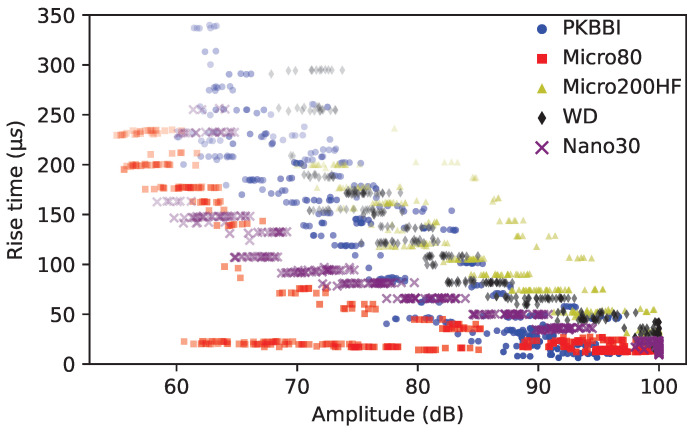
Rise time versus amplitude recorded by different sensors. The color density indicates the variation of the source–sensor distance: the lighter the color, the larger the source–sensor distance.

**Figure 12 sensors-24-02421-f012:**
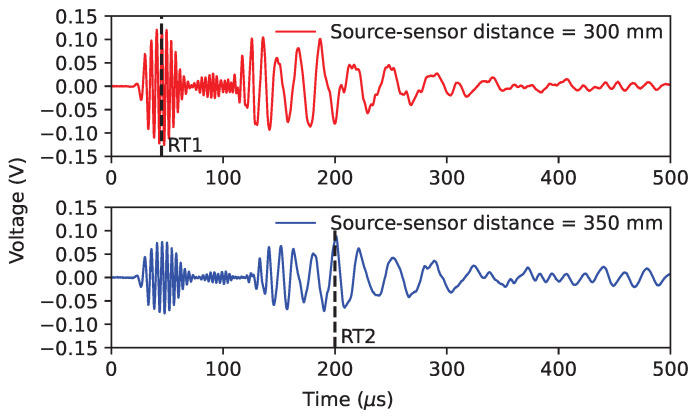
Waveforms recorded in the time domain by the Micro80 sensor at 300 mm and 350 mm. RT1 and RT2 indicate two rise times of these two signals.

**Figure 13 sensors-24-02421-f013:**
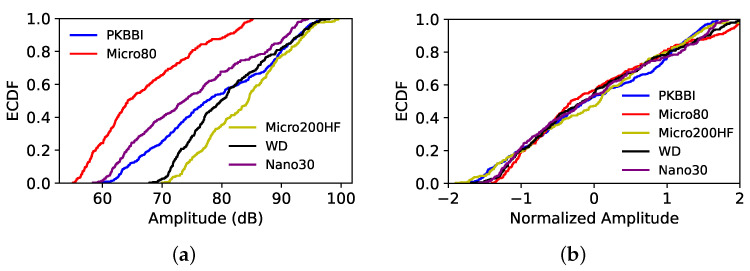
ECDF of the amplitude recorded by five sensors: (**a**) the raw data; and (**b**) the normalized data using the ZSN method.

**Figure 14 sensors-24-02421-f014:**
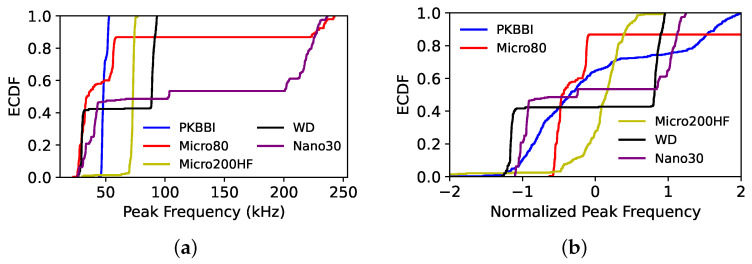
ECDF of the peak frequency recorded by five sensors: (**a**) the raw data; and (**b**) the normalized data using the ZSN method.

**Figure 15 sensors-24-02421-f015:**
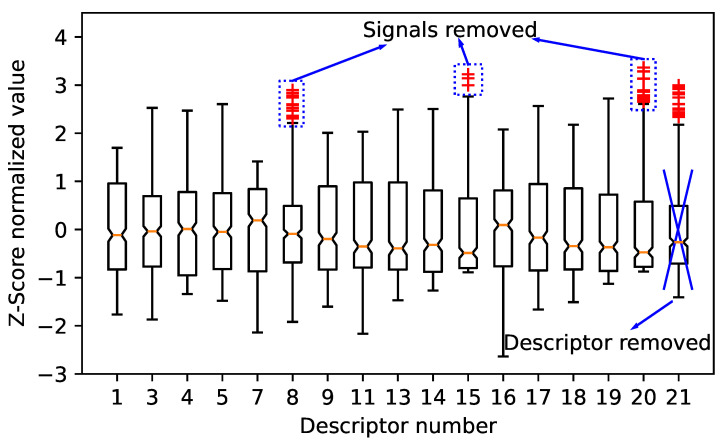
Boxplot of the Z-score normalized value of 17 descriptors recorded by the PKBBI sensor.

**Figure 16 sensors-24-02421-f016:**
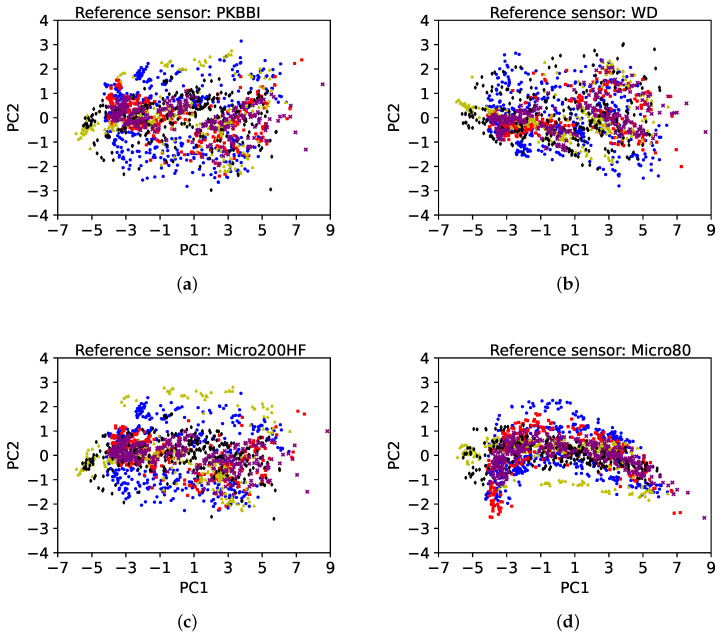
Two-dimensional scatter of the new datasets of the first three components (PC1, PC2) of five sensors. The reference eigenvector matrix is derived by the sensor: (**a**) the PKBBI sensor; (**b**) the WD sensor; (**c**) the Micro200HF sensor; (**d**) the Micro80 sensor; and (**e**) the Nano30 sensor.

**Figure 17 sensors-24-02421-f017:**
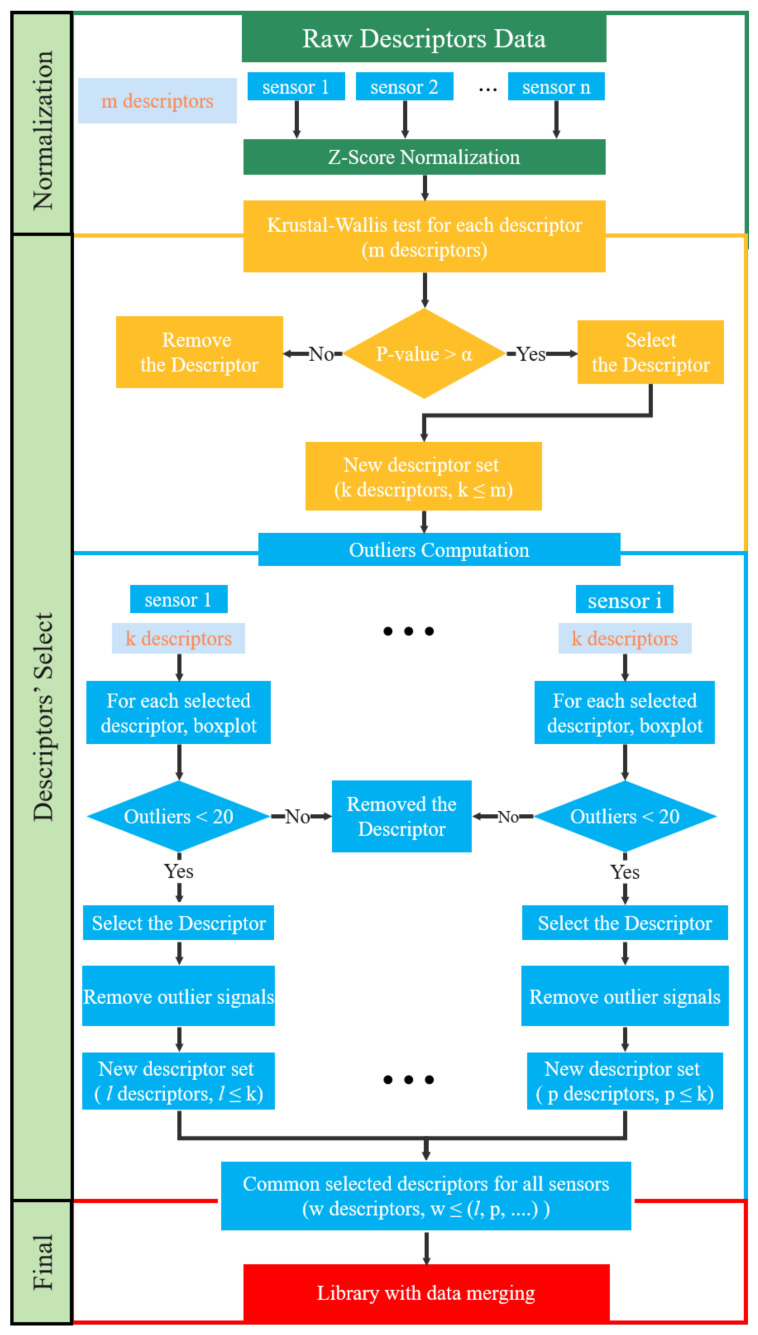
Flowchart showing the detailed steps of the normalization, the Kruskal–Wallis test, and the outlier computation to define a common subset of features for multiple sensors.

**Table 1 sensors-24-02421-t001:** Details of the five sensors, produced by the MISTRAS group.

Sensor	Resonant Frequency ^1^, Ref 1 V/(m/s) (kHz)	Resonant Frequency ^2^, Ref 1 V/µbar	Operating Frequency Range (kHz)	Sensor Diameter (kHz) ϕ (mm)
PKBBI	-	-	50–400	20
WD	125	450	125–1000	17.8
Micro200HF	2500	-	500–4500	9.5
Nano30	140	300	125–750	8
Micro80	250	325	200–900	10

According to ^1^ ASTM E 1106-96 [35]. ^2^ ASTM E 976-84 [36].

**Table 2 sensors-24-02421-t002:** Descriptor set of acoustic emission (AE) signals.

	Descriptor	Symbol	Unit
1	Amplitude	A	dB
2	Energy	E	V^2^
3	Zero-crossing rate	ZCR	-
4	Rise time	RT	μs
5	Temporal centroid	TC	μs
6	Temporal decrease	TD	${V\mu s}^{-1}$
7	Partial Power1 ]0, 100]	PP1	-
8	Partial Power2 ]100, 225]	PP2	-
9	Partial Power3 ]225, 300]	PP3	-
10	Partial Power4 ]300, 500]	PP4	-
11	Frequency centroid	FC	kHz
12	Peak frequency ]0, 1000]	PF	kHz
-	Peak frequency 1 ]0, 100]	PF1	kHz
-	Peak frequency 2 [100, 1000]	PF2	kHz
13	Spectral spread	SS	kHz
14	Spectral skewness	SSk	-
15	Spectral kurtosis	SK	-
16	Spectral slope	SSlope	${\mathrm{kHz}}^{-1}$
17	Roll-off frequency	-	kHz
18	Spectral spread to peak	SSP	kHz
19	Spectral skewness to peak	SSKP	-
20	Spectral kurtosis to peak	SKP	-
21	Roll-on frequency	-	kHz

**Table 3 sensors-24-02421-t003:** *p*-value obtained with the Kruskal–Wallis test for different normalization methods. (ZSN: Z-score normalization; MC: mean centered; PS: Pareto scaling; VSS: variable stability scaling; PT: power transformation; MMM: min–max normalization; MMADN: median and median absolute deviation normalization; VTN: Tanh normalization; LS: logistic sigmoid; HT: hyperbolic tangent).

Descriptor	ZSN	MC	PS	VSS	PT	MMN	MMADN	VTN	LS	HT
1. A	1.00	0.83	0.98	0.96	1.00	0.00	0.00	1.00	1.00	1.00
2. E	0.00	0.00	0.00	0.00	0.00	0.01	0.78	0.00	0.00	0.00
3. ZCR	0.70	0.12	0.39	0.46	0.63	0.00	0.00	0.70	0.70	0.70
4. RT	0.93	0.19	0.65	0.70	0.88	0.00	0.00	0.93	0.93	0.93
5. TC	0.97	0.54	0.89	0.46	0.96	0.00	0.00	0.97	0.97	0.97
6. TD	0.00	0.00	0.00	0.00	0.00	0.00	0.02	0.00	0.00	0.00
7. PP1	0.93	0.76	0.87	0.11	0.72	0.00	0.02	0.93	0.93	0.93
8. PP2	0.70	0.59	0.65	0.35	0.45	0.00	0.07	0.70	0.70	0.70
9. PP3	0.95	0.26	0.52	0.92	0.75	0.00	0.00	0.95	0.95	0.95
10. PP4	0.13	0.00	0.00	0.00	0.03	0.00	0.04	0.13	0.13	0.13
11. FC	0.56	0.01	0.14	0.44	0.64	0.00	0.03	0.56	0.56	0.56
12. PF	0.00	0.00	0.00	0.00	0.02	0.00	0.00	0.00	0.00	0.00
13. SS	0.85	0.00	0.00	0.00	0.00	0.00	0.09	0.85	0.85	0.85
14. SSK	1.00	0.83	0.88	0.89	0.89	0.00	0.00	1.00	1.00	1.00
15. SK	0.98	0.18	0.24	0.20	0.60	0.00	0.00	0.98	0.98	0.98
16. SSlope	0.68	0.02	0.43	0.37	0.77	0.00	0.00	0.68	0.68	0.68
17. Roll-off	0.87	0.00	0.16	0.00	0.46	0.00	0.45	0.87	0.87	0.87
18. SSP	0.95	0.00	0.06	0.07	0.61	0.00	0.47	0.95	0.95	0.95
19. SSKP	0.93	0.61	0.86	0.39	0.06	0.00	0.00	0.93	0.93	0.93
20. SKP	0.97	0.01	0.08	0.24	0.27	0.00	0.00	0.97	0.97	0.97
21. Roll-on	0.99	0.00	0.09	0.01	0.54	0.00	0.09	0.99	0.99	0.99

## Data Availability

The raw data supporting the conclusions of this article will be made available by the authors on request.

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
