# Peer review of "Reduction in the Sensor Effect on Acoustic Emission Data to Create a Generalizable Library by Data Merging"

_sensors, 2024, doi:10.3390/s24082421_

Round 1

Reviewer 1 Report

Comments and Suggestions for Authors

In the manuscript ‘Investigation and reduction of the sensor effect on acoustic emission signals recorded on PMMA plate: towards a library with data merging’, the authors present a research study on the effect of different sensors on the waveforms and related descriptors during AE tests. They try to explain the influence of the different sensors on the recorded waveforms and propose an approach for data merging independent of the sensor type. While the findings are interesting and they can contribute to the existing literature, it is necessary that the presentation of the results is significantly improved. More specific comments can be found below:

- Fig. 1: Were measurements performed on the other side of the plate as well to examine the influence of the thickness on the monitored waveforms?

- Line 127: What is the criterion for the choice of the specific acquisition parameters?

- Line 138: How are the values of 0.8 and 0.5 chosen?

- Fig. 3: It is not clear how this graph is obtained.

- Title 2.3: Should be postprocessing.

- Fig. 4: What are the units of magnitude?

- Section 3.1: While the results in the graphs of this section are interesting, the discussion should be better structured in order to explain the differences between the sensors with respect to the different types of waveforms. At the moment, the text is a bit chaotic.

- Fig. 9a: It is not clear what the three different ‘classes’ represent.

- Section 3.2: Similar comment as before. Due to the amount of information, it is difficult to follow the discussion. It should be better structured.

- A discussion section should be added at the end of Section 3 for a better summary of the main findings.

- As a general comment for further research: numerical simulations are necessary to support the experimental findings.

Comments on the Quality of English Language

The quality of the English language is quite good. However, the authors should closely go through the text and correct grammar and syntax errors to make it easier to read.

Reviewer 2 Report

Comments and Suggestions for Authors

This paper determines a procedure for combining multi-sensor data by merging data from different sensors, which measure the same AE sources. The sensors do not depend on each other, but they can be combined to give a more complete image of the AE source in the library. The authors propose a method to reduce the sensor effect. The acoustic emission sources are generated by the pencil-lead break (PLB) procedure on a specimen surface under controlled conditions to emit a wave that sensors can then record. The emitted frequency content of this source is controlled by the length and the orientation angle of the pencil lead break. The paper studies the influence of the sensor effect with different source-sensor distances on the characteristics of the AE signals, obtained by PLB on a 3 mm thick PMMA plate. By comparing the AE results, obtained by five different sensors, the influence of the sensor effect is illustrated. A procedure is proposed to reduce the sensor effect, based on the normalized data, the Krusal-Wallis test, outliers computation, and the Principal Component Analysis. Subset of features is chosen that can be used for all sensors.

Some shortcomings of the paper are the following:

1. (Line 197): m must be replaced by mi and σ must be replaced by σi.

2. (Inscription under Figure 5): The phrase “Waveforms of signals recorded by PKBBI sensor, Micro80 sensor, Micro200HF sensor, and WD sensor plate…” must be replaced by the phrase “Waveforms of signals recorded by PKBBI sensor, WD sensor, Micro200HF sensor, Micro80 sensor, and Nano30 sensor…”.

3. (Inscription under Figure 6): The phrase “FFT of signals recorded by PKBBI sensor, Micro80 sensor, Micro200HF sensor, and WD sensor plate…” must be replaced by the phrase “FFT of signals recorded by PKBBI sensor, WD sensor, Micro200HF sensor, Micro80 sensor, and Nano30 sensor …”

4. All references must be accompanied by doi, namely Refs. 30 – 39.

Reviewer 3 Report

Comments and Suggestions for Authors

Overall, this article has conducted a relatively comprehensive and in-depth study on the sensor effect of acoustic emission signal. The experimental design is reasonable and the analysis method is scientific. It has certain reference value for the selection and optimization of acoustic emission signal sensors. However, the article still has room for improvement in the following aspects:

1)
The title of the article is long and it is recommended to simplify it to make it more concise and accurately reflect the core content of the article. And the period in the title is not necessary.

2) The introduction of the article fully introduces the background of acoustic emission technology, but does not fully explain the importance of sensor effects in acoustic emission signal processing. It is recommended to add relevant background introduction to explain the impact of sensor effects on acoustic emission signal processing.

3) The experimental part of the article describes the impact of different sensors on acoustic emission signals, but it lacks integration with actual application scenarios. It is recommended to add some practical application cases to illustrate the impact of sensor effects on acoustic emission signal processing to make the article more practical.

4) When analyzing sensor effects, the article mainly focuses on time domain and frequency domain characteristics. It is recommended to add discussion of time-frequency domain characteristics to more fully analyze sensor effects.

5) The article uses statistical methods such as principal component analysis when proposing methods to reduce sensor effects. It is recommended that the principles and advantages of these methods be elaborated in more detail to provide readers with a better understanding.

Comments on the Quality of English Language

See comments and suggestions for authors

Reviewer 4 Report

Comments and Suggestions for Authors

The authors perform a thorough in-depth analysis of the various detector characteristics for AE analysis. Such a study is dearly needed because many practitioners use only the detectors which are available (very few) and rely on their analysis. While this is fine in most cases, the authors demonstrate that different detectors give different results in the time and frequency domain. Analysing data will allow a conclusion on the nature of the source.

The authors also provide an excellent flow chart to explain details steps which are taken during normalization of the AE signals, the Krustals-Wallis test and outliers computation. This can lead to automatisation which several groups are undertaking using machine learning techniques. I fully recommend the paper for publication.
